

# Solar FTIR measurements of NO$_x$ vertical distributions: Part I) First observational evidence for a seasonal variation in the diurnal increasing rates of stratospheric NO$_2$ and NO

Pinchas Nürnberg[1], Markus Rettinger[1], and Ralf Sussmann[1]

[1]Karlsruhe Institute of Technology, IMK-IFU, Garmisch-Partenkirchen, Germany

*Correspondence to*: Pinchas Nürnberg (pinchas.nuernberg@kit.edu)





**Abstract**

Observations of nitrogen dioxide ($NO_2$) and nitrogen oxide (NO) in the stratosphere are relevant to understand long-term changes and variabilities in stratospheric nitrogen oxide ($NO_x$) and ozone ($O_3$) concentrations. Due to the versatile role of $NO_2$ and NO in stratospheric $O_3$ photochemistry they are important for recovery and build-up of $O_3$ holes in the stratosphere, and therefore can indirectly affect the human life. Thus, we present in this work the evaluation of $NO_2$ and NO stratospheric partial columns (> 16 km altitude) retrieved from ground-based Fourier-transform infrared (FTIR) measurements from over 25 years at Zugspitze (47.42° N, 10.98° E, 2964 m a.s.l.) and 18 years at Garmisch (47.47° N, 11.06° E, 745 m a.s.l.), Germany. The obtained stratospheric columns are only weakly influenced by tropospheric pollution and show only a very small bias of (2.5±0.2) % when comparing $NO_2$ above Zugspitze and Garmisch. Stratospheric columns of both $NO_2$ and NO show a diurnal increase in dependence of local solar time (LST). We quantified this behavior by calculating diurnal increasing rates. Here, we find mean values for the $NO_2$ diurnal increasing rate of $(0.89\pm0.14)\cdot10^{14}$ $cm^{-2}$ $h^{-1}$ and $(0.94\pm0.14)\cdot10^{14}$ $cm^{-2}$ $h^{-1}$ at Zugspitze and Garmisch, respectively. The mean NO a.m. diurnal increasing rate above Zugspitze can be found to be $(1.42\pm0.12)\cdot10^{14}$ $cm^{-2}$ $h^{-1}$. Regarding the seasonal dependency of these increasing rates, for the first time, we were able to detect a significant seasonal variation of both $NO_2$ diurnal increasing rates and NO a.m. diurnal increasing rates experimentally with a maximum of $(1.13\pm0.04)\cdot10^{14}$ $cm^{-1}$ $h^{-1}$ for $NO_2$ and $(1.76\pm0.25)\cdot10^{14}$ $cm^{-1}$ $h^{-1}$ for NO in September and a minimum of $(0.71\pm0.18)\cdot10^{14}$ $cm^{-1}$ $h^{-1}$ in December for $NO_2$ and a minimum of $(1.18\pm0.41)\cdot10^{14}$ $cm^{-1}$ $h^{-1}$ in November for NO. This similar behavior may be explained by the interconnection of both species in stratospheric photochemistry. The outcome of this work is a retrieval and analyzation strategy of FTIR data for $NO_x$ stratospheric columns, which can help to further validate photochemical models or improve satellite validations. The first use of this data set is shown in a companion paper (Nürnberg et al., 2023) extracting experiment-based $NO_x$ scaling factors describing the diurnal increase out of the retrieved partial columns and validating recently published model-based scaling factors.



## 1 Introduction

Reactive nitrogen oxides ($NO_x$) as nitrogen monoxide (NO) and nitrogen dioxide ($NO_2$) play a crucial role in atmospheric photochemistry both in the troposphere and in the stratosphere (Crutzen, 1970). In the tropospheric boundary layer, the $NO_x$ origin is mainly anthropogenic from the combustion of fuels and the use of nitrogen-based fertilizers. To a lower extent, biomass burning and biological processes in soils contribute to $NO_x$ production (Crutzen, 1979). In the upper troposphere near the tropopause, $NO_x$ concentration is mainly controlled by lightning events and air traffic. As a precursor for several harmful

air pollutants, e.g. ozone ($O_3$) and nitric acid ($HNO_3$), the building of $NO_x$ in the troposphere directly affects human health (World Health Organization. Regional Office for Europe, 2003). In the stratosphere, $NO_x$ is produced by the photolysis of nitrous oxide ($N_2O$), which was transported through the tropopause and is a part of the biospheric nitrogen cycle (Johnston, 1992). As an important part of the $O_3$-destroying catalytic cycle $NO_x$ controls the abundance of $O_3$ in the stratosphere (Murphy et al., 1993). Consequently, since the Montreal Protocol was passed in 1987 with the aim to protect the stratospheric $O_3$ layer,

the monitoring of both $O_3$ and $NO_x$ became the focus of attention of much research (Tripp, 1987).

  The global distributions of atmospheric $NO_2$ and NO have been monitored by satellite missions since 1967 in various operational modes (Godin-Beekmann, 2010; Rusch, 1973): $NO_x$ data products are available from nadir-looking instruments like TROPOMI, GOME and SCIAMACHY (Griffin et al., 2019; Richter and Burrows, 2002; Sierk et al., 2006), limb-viewing instruments like MIPAS and OSIRIS (Funke et al., 2005; Haley et al., 2004) and from solar occultation measurements namely

ACE-FTS and SAGE III/ISS (Fussen et al., 2005; Chu et al., 2002). The validation and correction of these data with ground-based measurements is still an ongoing process which significantly reduced statistical and systematic errors between different satellite and ground-based measurements in the past decades (Van Geffen et al., 2022; Verhoelst et al., 2021; Kerzenmacher et al., 2008; Wetzel et al., 2007; Brohede et al., 2007; Heue et al., 2005). However, in comparing data (satellite vs. ground) which are in general recorded during different times of the day a major problem arises: There is a strong diurnal variation of

stratospheric $NO_x$ due to a complex photochemistry (Solomon et al., 1986), so biases arise just due to time mismatch.

  Facing this mismatch, a common method is the use of correction factors calculated from photochemical models to extrapolate retrieved data to the same time of the day. By now these models have a high accuracy giving information about $NO_x$ concentration in dependence of altitude, latitude and time of the day (Dubé et al., 2020; Strode et al., 2022). However, to the best knowledge of the authors, a reliable analysis of long-term observations of $NO_x$ stratospheric partial columns and their

diurnal variations, which could be used for validation of model data, does not exist. This is due to the lack of measurements able to record stratospheric $NO_x$ as function of the time of a day.

  For the ground-based observation of $NO_x$ several different measurement techniques are well established such as microwave radiometers (MR), zenith sky (ZS) and multi axis (MAX) differential optical absorption spectroscopy (DOAS) and Fourier-transform infrared (FTIR) spectroscopy. The MR technique is sensitive at high altitudes and offers the possibility to obtain

$NO_2$ columns independent of night- and daytime (Ricaud et al., 2004). ZS-DOAS or Système d'Analyse par Observations Zénithales (SAOZ) instruments are well established at many stations all over the globe, and provide long-term information about trace gas columns (e.g. $NO_2$) in the stratosphere (Platt and Stutz, 2008; Vandaele et al., 2005; Pommereau and Goutail, 1988; Solomon et al., 1987). However, these instruments especially have a good sensitivity at high SZA near sunrise or sunset (Tack et al., 2015). To get information at high SZA, MAX-DOAS measurements are performed providing information about

tropospheric trace gas concentrations at different times of the day (Dimitropoulou et al., 2020; Hönninger et al., 2004). However, these measurements do not provide information about trace gas concentrations near the tropopause and in the lower stratosphere.

  Accurate information on $NO_2$ and NO columns are accessible via FTIR solar absorption spectrometry, which can cover the whole diurnal variation of $NO_x$ (Fischer, 1993). Since the first ground-based FTIR measurements of $NO_2$ (Camy-Peyret et al.,

1983) and NO (Hanst et al., 1982), some effort was made in monitoring seasonal trends and diurnal variation of stratospheric and tropospheric $NO_x$ (Zhou et al., 2021; Yin et al., 2019; Virolainen et al., 2014; Hendrick et al., 2012; Flaud et al., 1988;





Rinsland et al., 1988). However, the multitude of this research investigated time periods covering only a few days up to several months. An examination of reliable long-term FTIR measurements with regard to stratospheric $NO_2$ columns was done by Hendrick et al. (2012). Even though the $NO_2$ diurnal variation is not discussed, the evaluation of 20 years of measurements

above Jungfraujoch depict a consistent picture of 1) the seasonal variability of stratospheric $NO_2$ columns which undergoes a maximum in summer and a minimum in winter and 2) a long-term trend which seems to show a slight decrease of stratospheric $NO_2$ in the order of 3.6 % over 20 years. Already before, a study of Sussmann et al. (2005) had quantified the $NO_2$ diurnal variation from ground-based FTIR measurements at the Zugspitze. This study successfully showed, that the stratospheric $NO_2$ diurnal variation can be measured at a high-altitude site without the strong influence of tropospheric pollution events

(Sussmann et al., 2005). However, due to the comparably short time period analyzed in this earlier study (2.5 years), a seasonal dependency could not be retrieved. Another reliable long-term study from Zhou et al. (2021) analyzed NO tropospheric and stratospheric partial columns retrieved from FTIR measurements above Xianghe and Maido. This study comprises both the seasonal variability of stratospheric NO with a maximum in winter and a minimum in summer and the diurnal variation of it in dependence of the local time (Zhou et al., 2021). However, a quantification regarding the seasonal dependence of the diurnal

increase was not discussed.

Therefore the goal of this work is i) to analyze the full Zugspitze and Garmisch FTIR time series covering more than 25 years (1995-2022) and 18 years (2004-2022) of measurements, respectively, to derive the diurnal increase of $NO_2$ and NO stratospheric columns above our mid-latitude sites while eliminating the impact of tropospheric pollution or tropopause variabilities, ii) investigate whether a significant seasonal variation of the $NO_2$ diurnal increase can be inferred, and iii) perform

a comparison to NO stratospheric columns to further validate the analyzation method and the reliability of the obtained data. The measurement data set published along with this paper will be a sound basis for validating current and upcoming photochemistry model simulations and improving satellite validation.

This paper is Part 1 of two companion papers dealing with the experimental description of the diurnal $NO_x$ variability above Zugspitze by means of ground-based FTIR measurements. Our paper will first discuss the stratospheric $NO_x$ photochemistry

and the consequences for the diurnal behavior of $NO_2$ and NO in Sect. 2. In Sect. 3 we will describe the retrieval strategy for $NO_2$ and NO from solar FTIR measurements at Zugspitze and Garmisch. Section 4 will focus on the retrieval results, the separation of the retrieved columns into stratospheric and tropospheric contributions, and the introduction of a pollution filter for the obtained stratospheric columns. The calculation of $NO_2$ diurnal increasing rates and their seasonal variation will be made in Sect. 5 followed by a comparison to NO a.m. diurnal increasing rates validating the analyzation method in Sect. 6.

Section 7 gives the summary and conclusions.



## 2 Photochemistry of stratospheric $NO_x$

As a background for our later FTIR-data interpretation, we thereafter present a short overview over the model understanding of $NO_x$ stratospheric photochemistry. More details can be found in the literature (Crutzen, 1970; Crutzen, 1979; Coffey et al., 1981; Cariolle, 1983; Jaeglé et al., 1994; Lary, 1997; Cohen and Murphy, 2003; Brasseur and Solomon, 2005).

During daytime the main $NO_2$ source is the photolysis of the reservoir species $HNO_3$ and $N_2O_5$, see Eq. (R1) and (R2), resulting in a consecutive increase of $NO_2$ within the day.

$$HNO_3 + hv \rightarrow NO_2 + OH, \tag{R1}$$
$$N_2O_5 + hv \rightarrow NO_2 + NO_3. \tag{R2}$$

Both reactions take place on a time scale of minutes to hours between sunrise and sunset and the kinetics depend on solar elevation.

The main NO source is the reaction of nitrous oxide ($N_2O$) with exited oxygen ($O(^1D)$) resulting from the photolysis of $O_3$ given in Eq. (R3) and (R4). This leads to a similar consecutive increase of NO within the day as seen for $NO_2$.

$$O_3 + hv \rightarrow O(^1D) + O_2, \tag{R3}$$
$$N_2O + O(^1D) \rightarrow 2\ NO. \tag{R4}$$

According to the model understanding, the reaction rate of Eq. (R1) to (R4) decreases after noon leading to a lower $NO_2$ and NO increase in the afternoon than observed in the morning.

Additionally, both $NO_x$ species are interconverted into each other very fast on time scales of seconds within the $O_3$-destroying nitrogen catalytic cycle

$$NO + O_3 \rightarrow NO_2 + O_2, \tag{R5}$$
$$NO_2 + O \rightarrow NO + O_2, \tag{R6}$$
$$\text{net: } O\ +\ O_3 \rightarrow 2\ O_2, \tag{R7}$$

and via the photolysis of $NO_2$ (Eq. (R8)), resulting in an equilibrium during daytime.

$$NO_2\ +\ hv \rightarrow NO + O. \tag{R8}$$

This equilibrium is reached very fast after sunrise and is nearly constant in the morning where the concentration increase of both species follows in a good approximation a linear behavior. In the afternoon, the equilibrium is changing due to the strong solar elevation dependency of Eq. (R8) and due to the increasing abundancy of $O_3$ with daytime (Wang et al., 2020; Strode et al., 2022). Consequently, after noon, the NO increase slows down, whereas $NO_2$ continues to increase. Between SZA = 80°-90° the trace gas concentrations are still influenced by the thermally driven reactions taking place at night, leading to a strong deviation from a linear behavior during very early morning.

## 3 FTIR measurement and retrieval strategy

### 3.1 Measurement

All data of this study are retrieved from long-term ground-based FTIR solar absorption measurements at the Zugspitze, Germany (47.42° N, 10.98° E, 2964 m a.s.l.) and Garmisch, Germany (47.47° N, 11.06° E, 745 m a.s.l.). The high-altitude observatory at Zugspitze is located in the German alps and can be treated as a clean site without strong influences from pollution events in the boundary layer. The observatory at Garmisch is located in direct vicinity to the Zugspitze, but 2219 m below in the countryside under the influence of urban pollution events from e.g. Munich. The used Bruker IFS 125HR spectrometers are operated continuously since 1995 at the Zugspitze and since 2004 at Garmisch. They operate with an actively controlled solar tracker and liquid-nitrogen cooled MCT (HgCdTe) and InSb detectors. Instrument and measurement details are described elsewhere in detail (Sussmann and Schäfer, 1997; Sussmann, 1999). The used data set for the Zugspitze comprises all available measurements since 1995 to now. Namely 19,552 spectra on 2,579 measurement days (7.58



measurements per measurement day on average) for the micro-window (MW) used for the $NO_2$ retrieval and 7,513 spectra on 2,247 measurement days (3.34 measurements per measurement day on average) for the NO retrieval. The maximum optical path difference is 175 cm and 250 cm, respectively. The used data set for Garmisch comprises all available measurements since 2004 to now. Namely 15,801 spectra on 2,114 measurement days (7.47 measurements per measurement day on average) for the MW used for the $NO_2$ retrieval.

## 3.2 Retrieval strategy

In this paper, $NO_2$ and NO volume mixing ratio (VMR) profiles and column amounts are derived from measured spectra using version 9.6 of the retrieval code PROFFIT (Hase et al., 2004). The used parameters of the two described retrievals are summarized in detail in Table S1 in the supplement. They are all optimized leading to minimum values of the resulting spectral residuals (measured minus calculated) and physically meaningful vertical VMR profiles. The main quality selection criterion after a successful retrieval (< 20 iterations) was a ratio of the noise-to-signal ratio (NSR) to the degrees of freedom for signal (DOFS) of $\frac{\text{NSR}}{\text{DOFS}} \leq 0.125$ for $NO_2$ and $\leq 0.2$ for NO, respectively. These settings have been determined by a tradeoff between data quality and data amount. The DOFS is a measure of the information content that can be attained on the vertical profile from the retrieval (Rodgers, 1998). Additionally, all spectra recorded at SZA > 80° were dropped because of the influence of the thermally driven reactions taking place at night, which can be dominant already near the terminator (SZA = 90°, see Sect. 2). The resulting mean calculated spectra for the $NO_2$ and the NO retrieval, their spectral residuals, and the NSR are shown in Fig. S1a and b and Fig. S2, respectively. The latter is NSR = 0.0694 % ($NO_2$) and 0.1603 % (NO) at the Zugspitze and 0.0631 % ($NO_2$) at Garmisch.

### 3.2.1 $NO_2$

For retrieval of $NO_2$ above the Zugspitze and above Garmisch a prominent infrared absorption line first suggested for atmospheric retrievals by Camy-Peyret et al. (1983) was used, utilizing a spectral MW ranging from 2914.3 cm$^{-1}$ to 2914.85 cm$^{-1}$. This MW includes a strong absorption of $CH_4$ at 2914.5 cm$^{-1}$ which is retrieved simultaneously. For both species ($NO_2$ and $CH_4$) we applied a simple first-derivative ($L_1$) smoothness constraint (Tikhonov, 1963). Vertical a priori profiles of the interfering species $H_2O$, $O_3$, $H_2CO$, OCS, and $C_2H_6$ were iteratively scaled within the retrieval. For $NO_2$ one single averaged a priori profile was taken from the Whole Atmosphere Community Climate Model (WACCM) version 6 generated by NCAR (Lamarque et al., 2013). Daily profiles from the GGG2020 software (Laughner et al., 2022) were used for the interfering species. The spectroscopy for all species is taken from ATMOS version 20200512 (Brown et al., 1996).

### 3.2.2 NO

For retrieval of NO above the Zugspitze the prominent doublet located at 1900.075 cm$^{-1}$ was used, utilizing a spectral MW ranging from 1899.900 cm$^{-1}$ to 1900.100 cm$^{-1}$. This MW was also used in previous studies (Zhou et al., 2021; Wiacek et al., 2006; Notholt et al., 1995). This MW includes an absorption line of $CO_2$ at 1899.995 cm$^{-1}$ which is retrieved simultaneously. For both species (NO and $CO_2$) we applied a $L_1$ Tikhonov regularization. A vertical a priori profile of $O_3$ was iteratively scaled within the retrieval. For the other interfering species $H_2O$ and $N_2O$ only a forward calculation was used along within retrieval of the other species. For NO one single averaged a priori profile was taken from the Whole Atmosphere Community Climate Model (WACCM) version 6 generated by the NCAR (Lamarque et al., 2013). Daily profiles from the GGG2020 software were used for the interfering species (Laughner et al., 2022). The spectroscopy for all species is taken from HITRAN2020 (Gordon et al., 2022).



### 4 NOₓ vertical profiles and pollution filter

Following the retrieval strategy and the quality control described in Sect. 3, $NO_x$ vertical profiles are derived above Zugspitze ($NO_2$ and NO) and above Garmisch ($NO_2$) for each spectrum and are shown in the supplement material in Fig. S3a (Zugspitze) and b (Garmisch) for $NO_2$ and in Fig. S4a and b for NO (red lines). From the remaining 16,023 (Zugspitze, $NO_2$), 14,460 (Garmisch, $NO_2$) and 6,213 (NO) spectra a mean DOFS of 1.38, 1.49 and 2.14, respectively, are derived.

### 4.1 Separation of the tropospheric and stratospheric column contributions

As mentioned in the introduction, one main issue of this work is the reduction of error sources influencing the reliability of the interpreted data. To avoid the influence of $NO_x$ variability in the troposphere and near the tropopause on the retrieved stratospheric $NO_x$ columns, in this section we will describe the separation of the derived columns into two partial columns, even though the obtained DOFS for the $NO_2$ retrieval are only 1.38 (Zugspitze) and 1.49 (Garmisch) and not 2.0. The lower partial column covers the troposphere and the lower stratosphere up to 16 km. The upper partial column covers the middle and

upper stratosphere above 16 km.

### 4.1.1 NOₓ partial column averaging kernels above Zugspitze and Garmisch

Figure 1a depicts the retrieved number density (mean over all measured spectra) of $NO_2$ as a function of altitude $z$ at Zugspitze (continuous gray line) and Garmisch (broken line) normalized to its maximum value in the stratosphere. Additionally, the partial column averaging kernels (PCK) for both retrievals below (red line) and above (blue line) 16 km altitude are shown.

For both stations, a nearly identical profile (gray) is obtained, confirming the retrieval method. The first local maximum extends over the lower troposphere up to 8 km altitude. This maximum reflects the mainly anthropogenic $NO_x$ sources in the boundary layer. Although the measurements are performed on a high-altitude site (Zugspitze), the influence of anthropogenic $NO_x$ sources from the boundary layer on the profile cannot be excluded. Another contribution certainly results from the a priori profiles (given the shape of the a prioris used as depicted in Fig. S3 (green line) along with the weak sensitivity of the PCK

< 16 km (red continuous line) for the 2.964-8 km range). Near the tropopause between 5 km and 15 km another local maximum is visible. This accumulation is typical for mid-latitudes and can be explained by mainly the influence of lightning in summer, the vertical transport of $NO_x$ from surface emissions and air traffic (Grewe et al., 2001). Above 16 km a large peak is apparent in the profiles with a maximum at ~ 30 km. Here, the stratospheric $NO_x$ / $O_3$ photochemistry is taking place which is the focus of this work.

Figure 1b depicts in the same manner the retrieved mean number density for NO normalized to its maximum value in the stratosphere against $z$ (gray line) and the PCK below (red line) and above (blue line) 16 km altitude. The NO profile (gray) shows analogous maxima as described above for $NO_2$. The lowest maximum results from anthropogenic emissions in the boundary layer, the maximum near the tropopause results from lightning events, vertical $NO_x$ transport and air traffic and the maximum at 30 km altitude reflects $NO_x$ / $O_3$ photochemistry in the stratosphere.

To give reason for a separation of the stratospheric columns from the lower ones, the PCK for < 16 km (red lines) and > 16 km (blue lines) altitude are depicted in Fig. 1a and b too.

The lower PCK of the $NO_2$ retrieval at Zugspitze (continuous red line, Fig. 1a) shows a moderate sensitivity in the altitude region between 2.964 km and 16 km with a maximum of 0.38 at 18 km. In contrast, the lower PCK of the $NO_2$ retrieval at Garmisch (dotted red line, Fig. 1a) shows a strong sensitivity in the lower altitude region with a maximum of 1.33 at 17 km.

However, for both retrievals the sensitivity of the lower PCK at high altitudes of 30 km is very low with 0.18 and 0.35, respectively. Here, both stratospheric PCK (blue line), which are very similar for the retrieval at Zugspitze (continuous line) and Garmisch (dotted line), show a high retrieval sensitivity of ca. 1 above 30 km and a comparably low sensitivity below 16 km.



For the NO retrieval at Zugspitze (Fig. 1b) a similar pattern is achieved. The sensitivity of the lower PCK (red line) is rather high above the tropopause with a maximum of 0.69 at 18 km but it decreases strongly to higher altitudes (0.11 at 30 km). In comparison, the stratospheric PCK (blue line) as seen for $NO_2$ shows a continuous high sensitivity to stratospheric variabilities with a value of ~ 1 above 30 km.

These findings make it reasonable to split up the obtained $NO_2$ and NO profiles into partial columns above and below 16 km altitude to avoid influences of variabilities near the tropopause and in the boundary layer upon the stratospheric partial column, although the resulting DOFS of the $NO_2$ retrieval are only 1.38 (Zugspitze) and 1.49 (Garmisch).

## 4.2 Pollution filter

In a next step the obtained $NO_x$ lower partial columns should be used to account for pollution events in the boundary layer which also could affect the data retrieved for the stratospheric partial column and especially their diurnal variability. Figure S5a-d in the supplement show the retrieved $NO_2$ partial columns above Zugspitze (top row) and above Garmisch (bottom row) below (left) and above (right) 16 km altitude in dependence of local solar time (LST) and partitioned into monthly data sets for the whole measurement period (blue to yellow symbols from January to December, see legend). To account for pollution events the evidently visible positive outliers of the lower partial columns (left) are identified via the interquartile range (IQR). All dates on which the retrieved lower partial column is above 1.5·IQR of the respective month are removed from the dataset and, consequently, will not show up in the stratospheric column too. The resulting pollution filtered $NO_2$ partial columns are shown in Fig. 2 for the measurements at Zugspitze (top row) and Garmisch (bottom row) and will be discussed in the next section. In the same manner we filtered the retrieved NO data set (see Fig. S6 top row (raw data) and bottom row (pollution filtered)) to account for tropospheric pollution events.

## 4.3 $NO_2$ partial columns above Zugspitze and Garmisch

In Fig. 2, the pollution filtered $NO_2$ partial columns below (left) and above (right) 16 km altitude measured at Zugspitze (top row) and Garmisch (bottom row) are shown. In comparison to the uncorrected data, the monthly data sets for both $NO_2$ partial columns are highly smoothened. In the troposphere and near the tropopause (lower partial column, Figure 2a and c) the $NO_2$ concentration does not show a diurnal variation in dependence of the LST. This behavior agrees with the literature and underlines the weak influence of photochemistry in the lower atmosphere (Li et al., 2021). Comparing the lower partial column above Zugspitze (Figure 2a) and Garmisch (Figure 2c), the difference in altitude (2219 m) of both observatories is directly visible. Due to the influence of anthropogenic emissions in the boundary layer, the lower partial column measured at Garmisch shows 7-10 times higher values than measured at Zugspitze, see also Fig. S7a in the supplement.

Contrary to this, both stratospheric partial columns (> 16 km) above Zugspitze (Figure 2b) and Garmisch (Figure 2d) have very similar values, see also for a direct comparison Fig. S7b. Due to the vicinity of both observatories it is to be expected that the stratospheric partial columns are practically identical. However, the question is whether the data retrievals can reflect this expectation because of the extremely differing station altitudes, with tropospheric $NO_2$ potentially impacting the Garmisch stratospheric retrievals more than in the Zugspitze case. When quantitatively comparing both timeseries, the mean bias of both partial columns over the whole period between 2004 and 2022 can be found to be only 2.5 %. The standard error of the bias is lower ($2 \cdot \sigma / \sqrt{(n)} = 0.28$ %), indicating that the 2.5 % difference between the stratospheric NO2 partial columns measured at Zugspitze and Garmisch is small but significant. Anyhow, the very low mean bias between both data sets validates the used retrieval method and confirms the data evaluation up to this point. Additionally, both stratospheric partial columns show a strong diurnal variation with LST. Here, the discussed diurnal increase from sunrise to sunset is well pronounced for every month. The influence of the stratospheric $NO_2$ seasonal cycle can be seen when comparing the different months (blue to yellow symbols from January to December, see legend). The $NO_2$ concentration in summer (greenish symbols) is ~3.5 times higher



than in winter time (blueish and yellowish symbols). This is in good agreement with long-term literature data from
Jungfraujoch, which is a high-altitude site at mid-latitudes (Hendrick et al., 2012).

## 5 NO₂ diurnal increasing rate

In this section we will use the pollution filtered NO₂ stratospheric partial columns measured at Zugspitze and Garmisch to
calculate diurnal increasing rates in dependence of the month. The latter quantitatively describes the seasonal variation in
diurnal stratospheric NO₂ concentrations. For validation of the observed behavior and the used retrieval method we will
furthermore correlate the obtained NO₂ diurnal increasing rates from both observatories (Zugspitze and Garmisch).

### 5.1 Calculation of monthly NO₂ diurnal increasing rates

Figure 3 shows the NO₂ stratospheric partial columns measured at Zugspitze (red open symbols) and Garmisch (blue closed
symbols) in dependence of the LST for every month. As discussed before, the data of both observatories are very similar when
comparing data of the same time of the day. Note that especially in winter, the data range measured at Garmisch is smaller due
to the combination of low solar altitude angle and the location of the observatory in the valley.
Within our observational data scatter, we cannot confirm from Fig. 3 any non-linear behavior of the NO₂ diurnal increase after
noon as forecasted from some models (Dubé et al., 2020; Mclinden et al., 2000). Instead, the measured NO₂ column appears
to increase linearly over the whole day for every time of the year. Consequently, we decided to extract NO₂ diurnal increasing
rates from the observed data by the determination of the slope of a linear fit over the whole day for every month at Zugspitze
(black dashed lines) and Garmisch (black dotted lines). A similar method for the determination of NO₂ diurnal increasing rates
was applied in earlier work (Sussmann et al., 2005; Li et al., 2021).
The results of the linear fits in dependence of the month are shown in Fig 4a for the measurements at Zugspitze (red open
symbols) and Garmisch (blue closed symbols). The calculated mean values are also indicated in the Figure and are
$(0.89\pm0.14)\cdot10^{14}$ cm$^{-2}$ h$^{-1}$ and $(0.94\pm0.14)\cdot10^{14}$ cm$^{-2}$ h$^{-1}$ for Zugspitze and Garmisch, respectively. The errors are two times the
standard error of the mean ($2\cdot\sigma/\sqrt{n}$), i.e., the mean values agree perfectly within error bars. Both increasing rates also agree
within error bars with the value of $(1.02\pm0.12)\cdot10^{14}$ cm$^{-2}$ h$^{-1}$ obtained in our earlier work for Zugspitze (Sussmann et al., 2005),
where a smaller data set (only 2 years) and a simpler retrieval approach had been utilized (using a total column retrieval with
a zero a priori below 10 km altitude instead of a full profile retrieval). Furthermore, Li et al. (2021) published for an even
smaller timespan (only one week in October 2018) a value of $(1.34\pm0.24)\cdot10^{14}$ cm$^{-2}$ h$^{-1}$ for the NO₂ diurnal increasing rate
above Table Mountain, California (34.38° N). This value roughly agrees with the values measured in this work for October
which are $(0.92\pm0.04)\cdot10^{14}$ cm$^{-2}$ h$^{-1}$ and $(1.01\pm0.05)\cdot10^{14}$ cm$^{-2}$ h$^{-1}$ for Zugspitze and Garmisch, respectively. Here, the even
smaller database but also the differing latitude (~13°) could explain the difference.
Besides the discussion of averaged NO₂ diurnal increasing rates and single monthly values, in Fig. 4a a clear seasonal
variability of the diurnal increasing rate obtained at Zugspitze and at Garmisch is visible. As reflected by the small error bars
of the calculated monthly mean values in Fig. 4a, for both observatories for the first time a seasonal cycle with a maximum of
$(1.13\pm0.04)\cdot10^{14}$ cm$^{-1}$ h$^{-1}$ in September and a minimum of $(0.71\pm0.18)\cdot10^{14}$ cm$^{-1}$ h$^{-1}$ in December can be shown
experimentally. For the quantitative validation of this new finding we directly correlate the obtained monthly NO₂ diurnal
increasing rates measured at Zugspitze and Garmisch in the next section, expecting both to have the same origin in stratospheric
photochemistry and therefore are correlated.

### 5.1.1 Correlation analysis of extracted diurnal increasing rates

Figure 4b shows the scatter plot of monthly NO₂ diurnal increasing rates measured at Garmisch against the ones measured at
Zugspitze. The error bars are $\pm2\cdot\sigma$ (standard deviation) from the linear fit. The red continuous line is the linear regression with





*x*- and *y*-error weighting with the method by York et al. (2004). With the assumption, that *x*- and *y*-errors are not correlated, the regression analysis results in the values given in Table 1. Additionally, the regression without error-weighting is shown

(red dotted line). Whereas the correlation coefficient *r* is independent of the errors, the *t*-value strongly depends on the error. If the *t*-value exceeds the critical *t*-value $t_{crit}(95\%) = 2.23$, a significant correlation within 95 % confidence is given. In this case, with a high correlation coefficient $r = 0.7899$ and with a *t*-value of 3.96 and 3.37 with and without error-weighting, respectively, it is very likely that the data are correlated. This result confirms that the shown seasonal variation of the $NO_2$ diurnal increasing rates is a real effect, which is probably originated in the stratospheric photochemistry at midlatitudes.

**6 NO increasing rate**

In this section we will analyze the retrieved NO stratospheric columns above Zugspitze. This analyzation is motivated by the question whether the observed seasonal dependence of the $NO_2$ diurnal increasing rate is originated in the stratospheric photochemistry and consequently can be seen in the NO data too.

**6.1 Calculation of monthly NO a.m. diurnal increasing rate**

Figure 5 shows the dependence of the stratospheric NO partial columns measured at Zugspitze (yellow open symbols) on the LST for every month. Unlike as seen for $NO_2$, for NO the non-linear behavior of the diurnal increase is well-pronounced and especially in summertime (mid row) after local solar noon the slope of the diurnal increase decreases significantly. As described in Sect. 2, this behavior can be attributed on the one hand to the strong solar elevation dependency of Eq. (R8). On the other hand, the increasing abundancy of $O_3$ with daytime influences the kinetics of Eq. (R7). Both effects lead to a change in the

chemical equilibrium between $NO_2$ and NO after local solar noon and explain the different afternoon behavior of both trace gases.

For the quantification of the diurnal increase of the NO stratospheric partial column and a good comparability to $NO_2$, here, we only make a monthly linear fit before noon (Figure 5, black dashed line) to extract a NO a.m. diurnal increasing rate for every month.

The results of the linear fits in dependence of the month are shown in Fig. 6 (yellow open symbols) together with the $NO_2$ diurnal increasing rates measured at Zugspitze (red open symbols). It can be seen, that the NO a.m. diurnal increasing rate shows a similar seasonal variation as the $NO_2$ diurnal increasing rate with a maximum of $(1.76\pm0.25)\cdot10^{14}$ cm$^{-1}$ h$^{-1}$ in September and a minimum of $(1.18\pm0.41)\cdot10^{14}$ cm$^{-1}$ h$^{-1}$ in November. Here, a correlation of $NO_2$ and NO diurnal increasing rates is likely.

However, the error bars of the linear fits of NO are significantly larger compared to $NO_2$. One main reason beside others for this effect is the smaller data base for the NO retrieval with less than one half of the spectra compared to $NO_2$ (16,023 vs. 6,213 spectra). This difference is originated in the use of another MW for the NO retrieval. Nevertheless, in the next section we will make a correlation analysis of both diurnal increasing rates to quantify the relationship between stratospheric $NO_2$ and NO.



### 6.1.1 NO₂-NO correlation analysis

In Fig. 6b a scatter plot of monthly NO a.m. diurnal increasing rates against the $NO_2$ diurnal increasing rates is shown, both measured at Zugspitze. The error bars are $\pm 2 \cdot \sigma$ from the linear fit. The red continuous line is the linear regression with *x*- and *y*-error weighting with method of York et al. (2004) as described in the Sect. 5.1.1. The red dotted line represents the fit without weighting. The results of the correlation analysis are given in Table 2.

    The high correlation coefficient of $r = 0.7798$ shows, that it is likely that the given data are related. Without considering the

error bars, the *t*-value (3.94) exceeds $t_{crit}$ (2.23) for a confidence level of 95 %, reflecting a significant correlation of the data within 95 % confidence. However, due to the larger error bars of the NO a.m. diurnal increasing rates, the application of error-weighting leads to an even smaller *t*-value of 0.83, resulting in no statistical correlation of both data sets within 95 % confidence. Nonetheless, we would like to argue, that the obvious similarity between the seasonality of the $NO_2$ and NO a.m. diurnal increasing rate observed in Fig. 6a is not accidental. If so, this observation would confirm our model understanding of

an interconnection of both trace gases in the stratospheric photochemistry.

### 7 Summary and Conclusions

    In this study, we analyzed long-term FTIR data recorded within the last 25 years at Zugspitze (47.42° N, 10.98° E, 2964 m a.s.l.) and Garmisch (47.47° N, 11.06° E, 745 m a.s.l.), Germany. We present a retrieval and analyzation strategy for the given FTIR data, which provides $NO_2$ and NO stratospheric partial columns (> 16 km altitude) which are only weakly

influenced by the tropospheric partial column and by pollution events. The obtained $NO_2$ stratospheric partial columns are with a bias of only (2.5±0.2) % very similar above Zugspitze and Garmisch, reflecting the reliability of the given analysis. The observed diurnal behavior of both the $NO_2$ and the NO stratospheric partial columns in dependence of the local solar time (LST) reflects the expected behavior described in the literature via photochemical model simulations: The $NO_2$ stratospheric partial column follows over the whole day and independent of the season a linear increase from sunrise to sunset. In a similar

way, the NO stratospheric partial column increases linearly before local solar noon. In the afternoon, the slope of the NO rise decreases significantly, especially in summertime. Beside these basic observations, we quantified the described diurnal increase of $NO_2$ and NO in dependence of LST by calculating monthly $NO_2$ diurnal increasing rates above Zugspitze and Garmisch with mean values of $(0.89\pm0.07) \cdot 10^{14}$ cm$^{-2}$ h$^{-1}$ and $(0.94\pm0.07) \cdot 10^{14}$ cm$^{-2}$ h$^{-1}$, respectively and monthly NO a.m. diurnal increasing rates above Zugspitze with a mean value of $(1.42\pm0.06) \cdot 10^{14}$ cm$^{-2}$ h$^{-1}$. Here, the mean $NO_2$ diurnal increasing

rates perfectly fits together with a literature value published by Sussmann et al. (2005). Additionally, for the first time we could show a significant seasonal variation of both the $NO_2$ diurnal increasing rate and the NO a.m. diurnal increasing rate experimentally with a maximum of $(1.13\pm0.04) \cdot 10^{14}$ cm$^{-1}$ h$^{-1}$ for $NO_2$ and $(1.76\pm0.25) \cdot 10^{14}$ cm$^{-1}$ h$^{-1}$ for NO in September and a minimum of $(0.71\pm0.18) \cdot 10^{14}$ cm$^{-1}$ h$^{-1}$ in December for $NO_2$ and a minimum of $(1.18\pm0.41) \cdot 10^{14}$ cm$^{-1}$ h$^{-1}$ in November for NO. Although the correlation analysis of both $NO_2$ and NO diurnal increasing rates give quantitative evidence for their

interconnection in the stratospheric photochemistry only within 85 % confidence, both diurnal increasing rates follow the same seasonal cycle.

    Part 2 of the companion papers (Nürnberg et al., 2023) will show the generalization of the observed $NO_x$ partial columns (> 16 km) by converting them into experiment-based $NO_x$ scaling factors describing the $NO_x$ diurnal variability in dependence of SZA, and will give a first comparison to recently published model-based scaling factors.

The data and analysis method given in this paper (Part 1) can be the first step for a latitude dependent (multi-station) data set reflecting the diurnal behavior of the stratospheric $NO_x$ column in dependence of season. Furthermore, the measurements with its high time resolution can serve as a basis for the validation of future photochemistry models and the improvement of satellite validation.



**Data availability**

The data underlying this publication can be obtained at any time from the corresponding author on demand.

**Competing Interests**

None.

**Acknowledgements**

Funding by the Federal Ministry of Education and Research of Germany within the Project ACTRIS-D (grant 01LK2001B)
is gratefully acknowledged. We acknowledge funding by the Helmholtz Research Program "Changing Earth – Sustaining our
Future" within the Research Feld "Earth and Environment" and by the KIT-Publication Fund of the Karlsruhe Institute of
Technology. We would like to thank Sarah A. Strode for carefully reading the manuscript.



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





**Table 1.** Results of Garmisch-Zugspitze diurnal increasing rate correlation analysis. The correlation coefficient $r$, $r^2$, and the calculated $t$-values from the linear regression with $x$- and $y$-error weighting and without. Significant correlation is achieved if the $t$-value exceeds the critical $t$-value $t_{crit}$ for the given confidence level.

| | Correlation coefficient $r$ | $r^2$ | $t$-value | $t_{crit}$(95 %) | Significant correlation within 95 % confidence? |
|---|---|---|---|---|---|
| with $x$- and $y$-error weighting | | | 3.96 | 2.23 | **yes** |
| | 0.7899 | 0.6239 | | | |
| no weighting | | | 3.37 | 2.23 | **yes** |

**Table 2.** Results of NO-NO$_2$ diurnal increasing rate correlation analysis. The correlation coefficient $r$, $r^2$ and the calculated $t$-values from the linear regression with $x$- and $y$-error weighting and without weighting. Significant correlation is achieved if the $t$-value exceeds the critical $t$-value $t_{crit}$ for the given confidence level.

| | Correlation coefficient $r$ | $r^2$ | $t$-value | $t_{crit}$(95 %) | Significant correlation within 95 % confidence? |
|---|---|---|---|---|---|
| with $x$- and $y$-error weighting | | | 0.83 | 2.23 | No |
| | 0.7798 | 0.6082 | | | |
| no weighting | | | 3.94 | 2.23 | Yes |

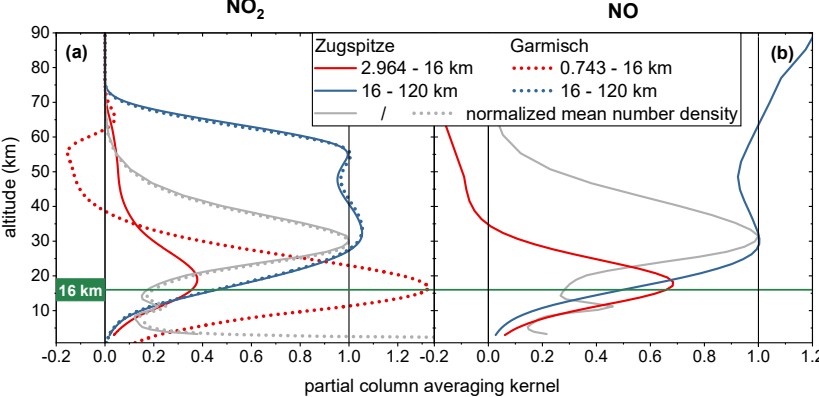

**Figure 1.** Retrieved partial column averaging kernels for below 16 km altitude (red lines) and above 16 km altitude (blue lines) of (a) NO₂ measured at Zugspitze (continuous lines) and Garmisch (dotted lines) and (b) NO measured at Zugspitze plotted in dependence of the altitude. Additionally, the respective normalized mean number density is shown in dependence of the altitude (gray lines). The green line indicates the splitting altitude 16 km.





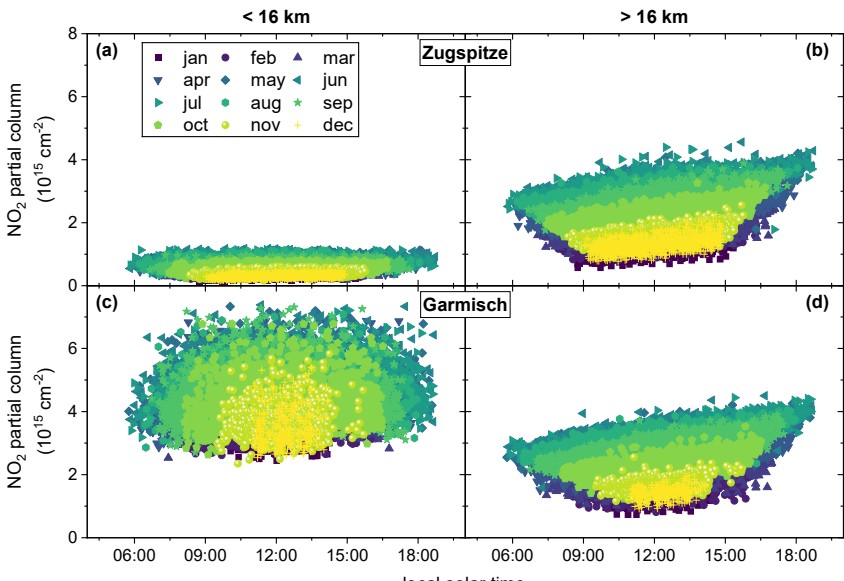

**Figure 2.** Retrieved pollution filtered NO₂ partial columns for every month below (a) and above (b) 16 km altitude measured at Zugspitze and below (c) and above (d) 16 km altitude measured at Garmisch in dependence of the local solar time (blue to yellow symbols from January to December, see legend).


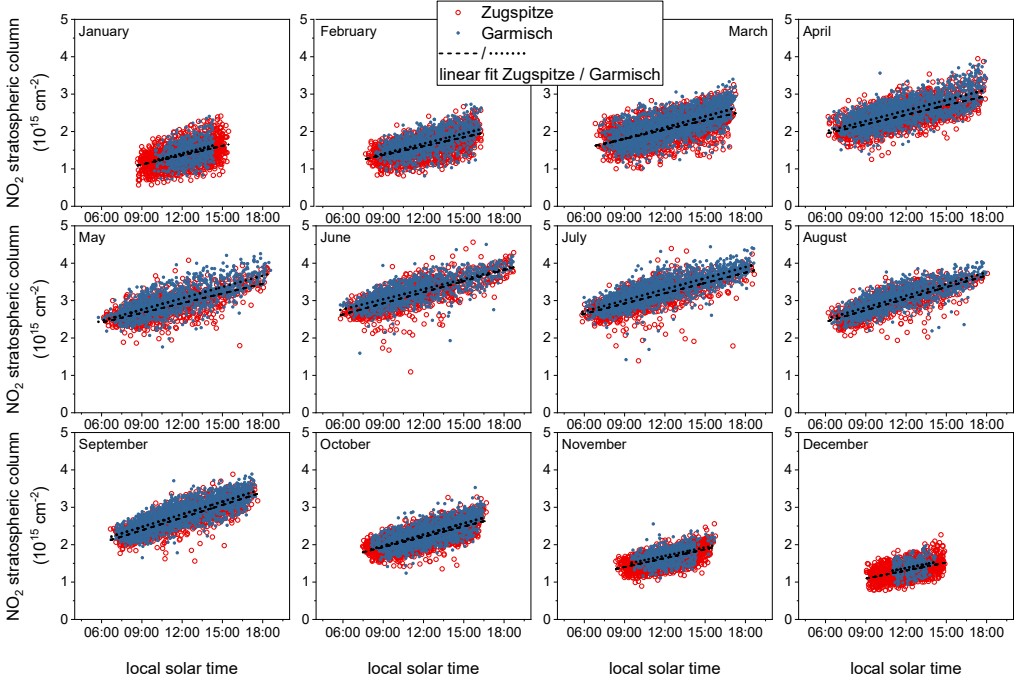

**Figure 3.** Retrieved pollution filtered NO₂ stratospheric columns (> 16 km) above Zugspitze (red open symbols) and Garmisch (blue closed symbols) for every month in dependence of the local solar time and linear fit between over the whole data range (black dashed and dotted lines for Zugspitze and Garmisch, respectively).






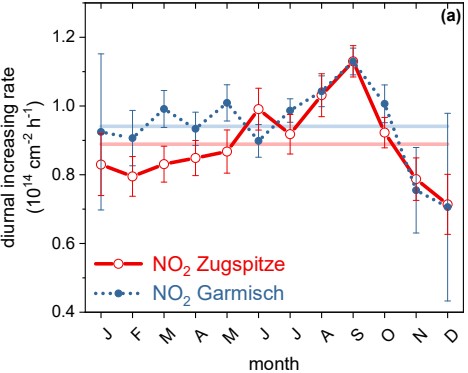 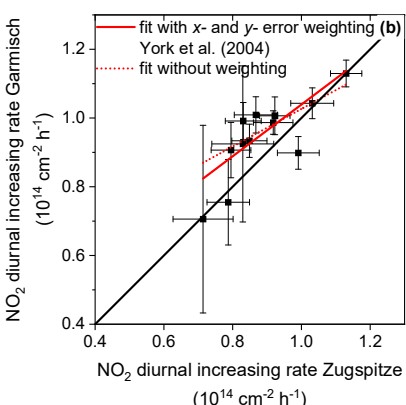

**Figure 4.** $NO_2$ diurnal increasing rates measured at Zugspitze and Garmisch. The error bars are $\pm 2 \cdot \sigma$ (standard deviation) from the linear fit. (a) Data for Zugspitze (red open symbols) and Garmisch (blue closed symbols) in dependence of the month. The lines are guides to the eye only. (b) Scatter plot of the data measured at Garmisch against the data measured at Zugspitze (black data points). Additionally, the linear regression with $x$- and $y$-error weighting with the method of York et al (2004) (red continuous line) and without weighting (red dotted line) is shown. The 1:1 line is given in black.

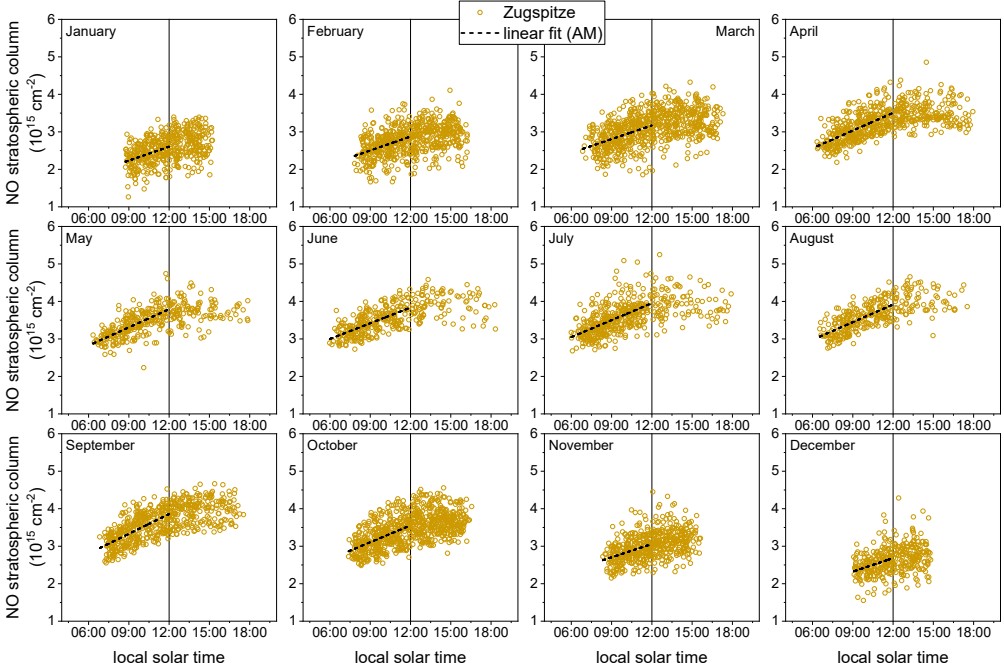

**Figure 5.** Retrieved pollution filtered NO stratospheric columns (> 16 km) above Zugspitze (yellow symbols) for every month in dependence of the local solar time and linear fit before local solar noon (black dashed line).





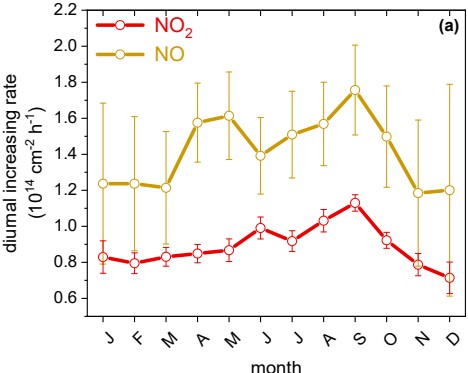
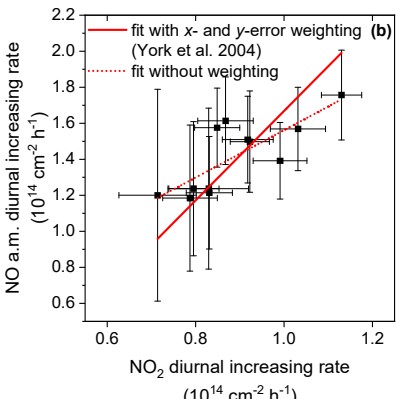

**Figure 6.** $NO_2$ diurnal increasing rates and NO a.m. diurnal increasing rates measured at Zugspitze. The error bars are $\pm 2 \cdot \sigma$ (standard deviation) from the linear fit. (a) Data for $NO_2$ (red symbols) and NO (yellow symbols) in dependence of the month. The lines are guides to the eye only. (b) Scatter plot of the NO data against the $NO_2$ data both measured at Zugspitze (black data points). Additionally, the linear regression with $x$- and $y$-error weighting with the method of York et al. (2004) (red continuous line) and without weighting (red dotted line) is shown.