# Peer review of "Solar FTIR measurements of $NO_x$ vertical distributions: Part I) First observational evidence for a seasonal variation in the diurnal increasing rates of stratospheric $NO_2$ and NO"

_EGUsphere, 2023_

## Referee Comment (RC1)

Review of Solar FTIR measurements of NOx vertical distributions: Part I) First observational evidence for a seasonal variation in the diurnal increasing rates of stratospheric NO2 and NO

This study examines stratospheric columns of NO2 and NO from FTIR measurements at two sites (Zugspitze and Garmisch). The retrieval process is discussed for each dataset. The results are then used to calculate the rate of change in both NO2 and NO as a function of local solar time for each month of the year. Overall, this paper is good. I look forward to part 2, which will be particularly helpful for validating model based diurnal scaling factors.

Questions & Comments

Line 45: Chu et al reference is for SAGE III/Meteor-3M, not SAGE III/ISS. The typical reference is

Cisewski, M., Zawodny, J., Gasbarre, J., Eckman, R., Topiwala, N., Rodriguez-Alvarez, O., ... & Hall, S. (2014, November). The stratospheric aerosol and gas experiment (SAGE III) on the International Space Station (ISS) Mission. In Sensors, Systems, and Next-Generation Satellites XVIII (Vol. 9241, pp. 59-65). SPIE.

Line 77: Which years does this trend correspond to?

Line 87: Clearly define what you mean by 'diurnal increase'

Line 155: Is it still possible to retrieve at SZA > 80? I understand that these values are not helpful for your diurnal increase calculation, but they would be very valuable for validating modelled NO2 and NO at sunrise and sunset. The photochemical model output at these times in highly uncertain, but necessary to use when considering measurements from occultation instruments.

Figures S3 and S4: red and green lines together will not pass the journal's colorblind test. I suggest changing the green line to black or blue.

Line 268: "The NO 2 concentration in summer (greenish symbols) is ~3.5 times higher than in winter time (blueish and yellowish symbols)"

- This is not very clear from the figure. I assume that the green and yellow symbols are on top of the blue and purple symbols, in which case it looks like the blue and green points have similar values. Perhaps it would be easier to see if you just chose a single colour for each season. Or else you could just refer to figure 3 instead as it more clearly shows the difference between the months.

Line 271: It is likely that your results do not show the non-linear behaviour because you are using a column measurement. Figure 1 of Dube et al 2021 shows that the slope and linearity of the NO2 diurnal cycle (from a model) varies considerably with altitude. This is probably worth mentioning.

Dubé, K., Bourassa, A., Zawada, D., Degenstein, D., Damadeo, R., Flittner, D., & Randel, W. (2021). Accounting for the photochemical variation in stratospheric NO2 in the SAGE III/ISS solar occultation retrieval. Atmospheric Measurement Techniques, 14(1), 557-566.

Some questions about Figure 4:

- Why does Zugspitze have a smaller slope in the first part of the year?
- Why do both stations show a steady increase in slope up to september and then a more rapid drop?
- Why does Garmisch have larger error in the winter?

Line 325: What are the other reasons?

Line 340: Are the changes in NO and NO2 consistent with one another? I think they should change in proportion to each other while in equilibrium (slope of scatter plot should follow 1:1 line)

Minor Edits

- In general, some of the wording and comma usage is strange. I suggest reading the manuscript through carefully.

Line 35: 'building' should be 'build-up'

Line 38: add a comma between 'cycle' and 'NOx'

Line 78: remove comma

Line 103: remove 'thereafter', change 'over' to 'of'

Line 107: change 'consecutive ' to 'continuous' Same on line 113.

Line 118: remove 'very fast'

Line 127: change 'daytime' to 'daylight'

Line 128: NO2 continues to increase at the same rate?

Line 241: change 'highly smoothened' to 'smooth'

Line 306: change 'This analyzation is motivated by the question whether' to 'This analysis is motivated by the question of whether'

Line 307, 327: change 'is originated in the' to 'originates in'

Line 310: change 'on the' to 'as a function of'

Line 314: change 'abundancy' to 'abundance'

Line 321: remove comma

Line 334: remove comma

Line 349: I do not understand this statement. The following line is also unclear: what is meant by the slope of the NO rise?

---

## Author Comment (AC1)

**Response to Anonymous Referees on acp-2023-1435**

**Solar FTIR measurements of NOx vertical distributions: Part I) First observational evidence for a seasonal variation in the diurnal increasing rates of stratospheric NO2 and NO**

We thank the Reviewers for their comments and suggestions. Below we provide our answers to their specific comments and the details of the changes made to the revised manuscript.

**Response to Anonymous Referee 1**

*Line 45: Chu et al reference is for SAGE III/Meteor-3M, not SAGE III/ISS. The typical reference is*
*Cisewski, M., Zawodny, J., Gasbarre, J., Eckman, R., Topiwala, N., Rodriguez-Alvarez, O., ... & Hall, S. (2014, November). The stratospheric aerosol and gas experiment (SAGE III) on the International Space Station (ISS) Mission. In Sensors, Systems, and Next-Generation Satellites XVIII (Vol. 9241, pp. 59-65). SPIE.*

    Done

*Line 77: Which years does this trend correspond to?*

    Text in line 77/78 of the revised manuscript has been modified accordingly:

    "2) a long-term trend which seems to show a slight decrease of stratospheric NO2 in the order of 3.6 % over 20 years from 1990 - 2010."

*Line 87: Clearly define what you mean by 'diurnal increase*

    Text in line 87-93 of the revised manuscript has been modified accordingly:

    "Therefore the goal of this work is i) to analyze the full Zugspitze and Garmisch FTIR time series covering more than 25 years (1995-2022) and 18 years (2004-2022) of measurements, respectively, to derive the slope of the linear fit of $NO_2$ and NO stratospheric columns in dependence of the local solar time (LST) - namely the diurnal increase - above our mid-latitude sites while eliminating the impact of tropospheric pollution or tropopause variabilities,"

*Line 155: Is it still possible to retrieve at SZA > 80? I understand that these values are not helpful for your diurnal increase calculation, but they would be very valuable for validating modelled NO2 and NO at sunrise and sunset. The photochemical model output at these times is highly uncertain, but necessary to use when considering measurements from occultation instruments.*

    We thank Reviewer #1 for this comment.

    It is possible to retrieve values at SZA > 80°. This data is available. However, this data is also uncertain due to the high influence of refraction at high SZA and consequently a bigger error on this data.

Text in line 158 of the revised manuscript has been modified accordingly:

"However, this dropped data is available from the corresponding author upon request."

*Figures S3 and S4: red and green lines together will not pass the journal's colorblind test. I suggest changing the green line to black or blue.*

Done

*Line 260: "The NO 2 concentration in summer (greenish symbols) is ~3.5 times higher than in winter time (blueish and yellowish symbols)"*

*This is not very clear from the figure. I assume that the green and yellow symbols are on top of the blue and purple symbols, in which case it looks like the blue and green points have similar values. Perhaps it would be easier to see if you just chose a single colour for each season. Or else you could just refer to figure 3 instead as it more clearly shows the difference between the months.*

Text in line 259-261 of the revised manuscript has been modified accordingly:

"The $NO_2$ concentration in summer (greenish symbols) is ~3.5 times higher than in winter time (blueish and yellowish symbols) which can be clearly seen when comparing summer and winter months in **Fehler! Verweisquelle konnte nicht gefunden werden.**."

*Line 271: It is likely that your results do not show the non-linear behaviour because you are using a column measurement. Figure 1 of Dube et al 2021 shows that the slope and linearity of the NO2 diurnal cycle (from a model) varies considerably with altitude. This is probably worth mentioning.*
*Dubé, K., Bourassa, A., Zawada, D., Degenstein, D., Damadeo, R., Flittner, D., & Randel, W. (2021). Accounting for the photochemical variation in stratospheric NO2 in the SAGE III/ISS solar occultation retrieval. Atmospheric Measurement Techniques, 14(1), 557-566.*

Text in line 259-261 of the revised manuscript has been modified accordingly:

"Within our observational data scatter, we cannot confirm from Fig. 3 any non-linear behavior of the $NO_2$ diurnal increase after noon as forecasted from some models (Dubé et al., 2020; Mclinden et al., 2000). Instead, the measured $NO_2$ column appears to increase linearly over the whole day for every time of the year. One reason for this deviation can be the altitude-dependence of the non-linearity of the $NO_2$ concentration discussed by Dubé et al. (2021), which cannot be addressed with $NO_2$ column data available in this work. However, we decided to extract $NO_2$ diurnal increasing rates from the observed data by the determination of the slope of a linear fit over the whole day for every month at Zugspitze (black dashed lines) and Garmisch (black dotted lines)."

*Some questions about Figure 4:*
*- Why does Zugspitze have a smaller slope in the first part of the year?*

We thank Reviewer #1 for this comment.

Only in March, the diurnal increasing rate of $NO_2$ above Zugspitze is significantly lower within the margin of error than above Garmisch. Therefore, we treat this data point as an outlier.

*- Why do both stations show a steady increase in slope up to september and then a more rapid drop?*

We thank Reviewer #1 for this comment.

This question we also thought about. But yet, we do not have a satisfactory answer. The origin of the observed seasonal effect can be various and should be the topic of further research.

*- Why does Garmisch have larger error in the winter?*

We thank Reviewer #1 for this comment.

This effect can be explained by the smaller data base due to the to the combination of low solar altitude angle and the location of the observatory in the valley compared to the Zugspitze as mentioned in line 272/273.

Text in line 272-274 of the revised manuscript has been modified accordingly:

"Note that especially in winter, the data range measured at Garmisch is smaller due to the combination of low solar altitude angle and the location of the observatory in the valley, leading to a higher uncertainty of the resulting data in the winter compared to Zugspitze."

*Line 329: What are the other reasons?*

We thank Reviewer #1 for this comment.

Here, we are aware of one reason, which should be the most important. However, we can not exclude others. That is why we mentioned only the main reason.

*Line 340: Are the changes in NO and NO2 consistent with one another? I think they should change in proportion to each other while in equilibrium (slope of scatter plot should follow 1:1 line)*

We thank Reviewer #1 for this comment.

Without taken model simulations into account, we cannot verify or refute this Assumption.

*Minor Edits:*

*Line 35: 'building' should be 'build-up'*
*Line 38: add a comma between 'cycle' and 'NOx'*

*Line 78: remove comma*
*Line 103: remove 'thereafter', change 'over' to 'of'*
*Line 107: change 'consecutive ' to 'continuous' Same on line 113.*
*Line 118: remove 'very fast'*
*Line 127: change 'daytime' to 'daylight'*
*Line 241: change 'highly smoothened' to 'smooth'*
*Line 306: change 'This analyzation is motivated by the question whether' to 'This analysis is motivated by the question of whether'*
*Line 307, 327: change 'is originated in the' to 'originates in'*
*Line 310: change 'on the' to 'as a function of'*
*Line 314: change 'abundancy' to 'abundance'*
*Line 321: remove comma*
*Line 334: remove comma*

Done

*Line 128: NO2 continues to increase at the same rate?*

Text in line 130 of the revised manuscript has been modified accordingly:

"Consequently, after noon, the NO increase slows down, whereas NO2 continues to increase with a similar rate."

*Line 354: I do not understand this statement. The following line is also unclear: what is meant by the slope of the NO rise?*

Text in line 354-355 of the revised manuscript has been modified accordingly:

"In the afternoon,  the increase in NO stratospheric partial column slows down significantly, especially in summertime."

**Response to Anonymous Referee 2**

*Line 34: Please, include some references about the lighting and air traffic controlling the NOx concentration in the upper troposphere.*

Done

*Line 64: MAX-DOAS measurements generally obtain information for lower SZA (compared to the high SZA of the twilights).*

Text in line 65 of the revised manuscript has been modified accordingly:

To get information at  lower SZA, MAX-DOAS measurements are performed providing information about tropospheric trace gas concentrations at different times of the day

*Lines 66-67: That is not entirely true, if the Free troposphere is considered to be representative of concentrations between the Boundary layer and the tropopause. In fact, by applying the method described in (Gomez et al., 2014), the Free Troposphere NO2 concentration can be estimated from MAX-DOAS measurements performed at mountain stations. In (Gil et al., 2015), for instance, that method was applied to Izaña MAX-DOAS data carried out over 3 years to study the seasonal evolution of NO2.*

We thank Reviewer #2 for this comment.

To make clear what we mean, we change the text accordingly:

However, these measurements do not provide information about trace gas concentrations  at the tropopause and in the lower stratosphere.

*Section 3.2: What temperature and pressure vertical profiles are used in the model?*

We thank Reviewer #2 for this comment.

As described in the supplement, the T and p profiles are taken from the National Centers for Environmental Prediction (NCEP)

*Line 194: Please, explain how are the partial column averaging kernels obtained.*

Text in line 196 of the revised manuscript has been modified accordingly:

"Additionally, the partial column averaging kernels (PCK, sum of the rows of the averaging kernel matrix over the respective altitude range of the partial column of interest) for both retrievals below (red line) and above (blue line) 16 km altitude are shown."

*Section 4.2: How often are these pollution outliers observed out of the studied cases? It would be interesting to study also the high pollution episodes and how these tropospheric events affect the stratosphere.*

We thank Reviewer #2 for this comment.

The analyzation of the observed outliers and therefore the study of pollution events is not part of this work but it is a very interesting topic to have a deeper look into.

*Section 6.1: How do you explain the difference of the NO2 seasonal evolution observed at both stations between April and June? (Figure 6).*

We thank Reviewer #2 for this comment.

In this section, we discuss the different of $NO_2$ and NO diurnal increasing rates. Fig. 6 shows only the different of both species, not of the stations.

*Technical Corrections:*

*Page 91: Do you mean "solid" instead "sound"?*

Text in line 65 of the revised manuscript has been modified accordingly:

The measurement data set published along with this paper will be a  solid basis for validating current and upcoming photochemistry model simulations and improving satellite validation.

*Page 112: "excited" instead of "exited".*

Done

*Line 139: MCT meaning.*

As described this stands vor HgCdTe (Mercury Cadmium Telluride)

*Line 192: "1a" instead of "1 a".*

Done

*Line 217: Something is missing in "of ca. 1"?*

The partial column averaging kernel do not have a unit.

*Line 248: you could mention the horizontal distance in km to show clearly how close the stations are.*

Text in line 248 of the revised manuscript has been modified accordingly:

"Due to the vicinity of both observatories (ca. 10 km) it is to be expected that the stratospheric partial columns are practically identical."

---

## Referee Report (RR1)

**Response to Authors on acp-2023-1435**

**Solar FTIR measurements of NOx vertical distributions: Part I) First observational evidence for a seasonal variation in the diurnal increasing rates of stratospheric NO2 and NO**

I thank the Authors for reviewing their manuscript and for taking into account the comments and suggestions of both referees. The manuscript has improved and most of the referees doubts have been clarified. I think that this paper should be publish in AMT. However, I think that some questions remain unclear.

I apologize to the authors for my confusion with figure 6. I actually have a question about figure 4. I am very sorry, but I would appreciate it if you could answer it.

My question about Figure 4 is similar to that of Referee 2. Why do you observe this "shift" between both stations in the first part of the year (from January to June)? Not only the diurnal increase rates of March are different, taking into account the error bars, but also in May. Do you have some explanation?

After reviewing the latest version of the manuscript, I also have a new comment. Lines 261-262: From Figure 3, I do not see that the NO2 concentration in summer is ~3.5 times the winter concentration. If the NO2 average concentration in winter around noon (for instance), is 1.5E15 cm-2, the concentration for summer months around noon is around 3-3.5 E15 cm-2. That means that the concentration in summer is about 2 times (twice) the concentration during winter.

---

## Referee Report (RR2)

Thank you for the revised manuscript. There are still a few points that need to be clarified:

Line 260/Figure 2: The way this is worded is still unclear as it now sounds like the green/yellow points are in Figure 3, rather than figure 2. Please rephrase this sentence.

Figure 4: Even though March is the only month when the two stations do not have diurnal increasing rates that agree within error, there is clearly a bias between the rates from January to May, that does not exist during the rest of the year. It is worth discussing possible reasons for this bias. In addition, if you think that March is an outlier, is the problem with Zugspitze or with Garmisch, and why?

Line 332: If you are only going to mention the main reason the text should probably not say "beside others"